Remote sensing-based detection of brown spot needle blight: a comprehensive review, and future directions

Singh Swati SZS0364@AUBURN.EDU
http://orcid.org/0000-0002-6125-7649 Narine Lana L. lln0005@auburn.edu
http://orcid.org/0000-0002-0176-1878 Willoughby Janna R.
Eckhardt Lori G.
College of Forestry, Wildlife and Environment, Auburn University, Auburn , AL , USA
Brygadyrenko Viktor
Electronic publication date: 2025 May 22
Publication date: 2025
Volume: 13
Electronic Location ID: e19407
Received 2024 Nov 19; Accepted 2025 Apr 10
Copyright: © 2025 Singh et al.
Copyright year: 2025
Copyright holder: Singh et al.
License: This is an open access article distributed under the terms of the Creative Commons Attribution License, which permits unrestricted use, distribution, reproduction and adaptation in any medium and for any purpose provided that it is properly attributed. For attribution, the original author(s), title, publication source (PeerJ) and either DOI or URL of the article must be cited.
License URL: https://creativecommons.org/licenses/by/4.0/

Keywords: Pine forestry, Needle disease, Lecanosticta acicola, Remote sensing, Geospatial analysis, Forest health monitoring

Funding: U.S. Forest Service 22-CA-11330160-067 This research was supported by funding from the U.S. Forest Service, under Award Number 22-CA-11330160-067. The funders had no role in study design, data collection and analysis, decision to publish, or preparation of the manuscript.

==============================
Pine forests are increasingly threatened by needle diseases, including Brown Spot Needle Blight (BSNB), caused by Lecanosticta acicola. BSNB leads to needle loss, reduced growth, significant tree mortality, and disruptions in global timber production. Due to its severity, L. acicola is designated as a quarantine pathogen in several countries, requiring effective early detection and control of its spread. Remote sensing (RS) technologies provide scalable and efficient solutions for broad-scale disease surveillance. This study systematically reviews RS-based methods for detecting BSNB symptoms, assessing current research trends and potential applications. A comprehensive bibliometric analysis using the Web of Science database indicated that direct RS applications for BSNB remain scarce. However, studies on other needle diseases demonstrated the effectiveness of multisource RS techniques for symptom detection, spatial mapping, and severity assessment. Advancements in machine learning (ML) and deep learning (DL) have further improved RS capabilities for automated disease classification and predictive modeling in forest health monitoring. Climate-driven factors, such as temperature and precipitation, regulate the distribution and severity of emerging pathogens. Geospatial analyses and species distribution modeling (SDM) have been successfully applied to predict BSNB pathogen’s range expansion under changing climatic conditions. Integrating these models with RS-based monitoring enhances early detection and risk assessment. However, despite these advancements, direct RS applications for BSNB detection remain limited. This review identifies key knowledge gaps and highlights the need for further research to optimize RS-based methodologies, refine predictive models, and develop early warning systems for improved forest management.

Introduction

Forest disturbances, such as insect infestations, disease outbreaks, and climate change, drive pathogen spread and threaten forest health, requiring advanced monitoring and mitigation (Dudney et al., 2021; Hartmann et al., 2022; Roshani et al., 2022). Pine species (Pinus spp.) are predominantly distributed across the Northern Hemisphere and hold significant economic and ecological value. They have also been introduced to temperate and subtropical regions of the Southern Hemisphere for timber production and ornamental use (Costanza et al., 2018; Ryu et al., 2018). However, pine forests worldwide face increasing threats from needle diseases, which are caused by fungal and oomycete pathogens that directly infect, and damage conifer foliage (Wingfield et al., 2015; Drenkhan et al., 2016).

Among these, Dothistroma Needle Blight (DNB or Red Band) is one of the most destructive, affecting over 80 pine species, with Corsican Pine (P. nigra J.F. Arnold subsp. laricio (Poir.)) being particularly susceptible. This disease is caused by the hemibiotrophic fungus Dothistroma septosporum or D. pini (sexual stage: Mycosphaerella pini) (Watt et al., 2009; Mullett, Peace & Brown, 2016). Similarly, Lophodermium Needle Cast, caused by Lophodermium seditiosum, primarily affects Scots Pine (P. sylvestris) (Jansons et al., 2024). Red Needle Cast (Phytophthora pluvialis) has emerged as a significant threat to pine and Douglas fir plantations in both the Northern and Southern Hemispheres, particularly in regions with high precipitation. This disease has been increasingly reported in radiata pine (P. radiata) and Douglas fir (Pseudotsuga menziesii), where it causes needle browning, premature defoliation, and reduced growth rates (Dick et al., 2014; Gomez-Gallego et al., 2019).

Additionally, Pine Needle Rust (Coleosporium asterum) infects various pine species (P. nigra, P. banksiana, P. resinosa, P. ponderosa, P. mugo, P. sylvestris) as well as plants in the Asteraceae family (University of Minnesota Extension). Black Spot Needle Blight, caused by Pestalotiopsis neglecta, is common in P. sylvestris var. mongolica (Ma et al., 2024). Brown Spot Needle Blight (BSNB), caused by Lecanosticta acicola, primarily affects Longleaf Pine (P. palustris) and Loblolly Pine (P. taeda) (Cunningham, 2022; Meinecke et al., 2024b).

Since 2016, outbreaks of needle diseases have been increasingly reported in the southeastern United States, particularly affecting loblolly pine (P. taeda), a key timber species in the region (Meinecke et al., 2024a). Symptoms initially manifest as small, irregular yellow spots that develop into larger, dark orange to brown lesions, often resin-soaked and encircled by a yellow halo (Mullett et al., 2018; Aglietti et al., 2021). The causal agent of these outbreaks has been identified as the native fungal pathogen L. acicola (Thum.) Syd. (formerly Mycosphaerella dearnessi, syn. Scirrhia asicola), responsible for BSNB foliar disease (Meinecke et al., 2024a).

Historically, BSNB foliar disease has not posed a major threat to P. taeda; however, the increasing scale and severity of recent outbreaks raise significant concern. Initially recognized in the southeastern United States, BSNB has severely impacted P. palustris, particularly in Christmas tree farms (Skilling & Nicholls, 1974). By the 1980s, timber losses caused by BSNB exceeded 453,000 cubic meters annually in southern pine species, including P. palustris (Cordell, Anderson & Kais, 1990). Over time, BSNB has emerged as a global concern, with invasive populations reported in many countries (EPPO, 2024b). The disease exhibits a broad host range, affecting over 70 taxa, primarily within the Pinus genus, including species of Cedrus and Picea (Tubby et al., 2023).

The southeastern United States, which accounts for 17% of global timber production is critical in national forestry (FAO Global Forest Resources Assessments, 2015). This region comprises 61% of the nation’s planted forests and 57% of its total wood volume, emphasizing the economic importance of key pine species such as loblolly pine (P. taeda), shortleaf pine (P. echinata), and slash pine (P. elliottii) (Oswalt et al., 2014; Fagan et al., 2018). States such as Alabama, Mississippi, Louisiana, and Arkansas are particularly reliant on these timber species, making them vulnerable to emerging threats such as BSNB (Datta, 2021; Meinecke et al., 2024a). In 2022, the Alabama Forestry Commission reported BSNB outbreaks in 36 of the state’s 67 counties, with wet summers and mild winters creating conditions favorable for L. acicola infection (U.S. Forest Service, 2022). This increasing prevalence highlights the need for improved strategies to mitigate its impact. As the disease progresses, it elevates stress susceptibility in trees, potentially leading to mortality with climatic factors playing a significant role in its spread (Ogris et al., 2023). Similar concerns exist globally with Dothistroma needle blight, which affects pine forests in western Canada and Fennoscandia (Drenkhan et al., 2016). The ecological and epidemiological parallels between Dothistroma species and L. acicola suggest that BSNB could expand its geographic range and pose a broader threat (Raitelaitytė et al., 2023). With changing climate conditions, proactive measures are needed to protect forestry resources on regional and global scales.

Remote sensing (RS) technologies have emerged as effective tools for identifying and monitoring needle disease symptoms such as Dothistroma needle blight. Studies using aerial and field-based spectroscopy imaging have been successfully applied to identify disease symptoms and assess severity levels (Coops et al., 2003; Watt et al., 2021; Watt et al., 2023). RS-based methods often involve the utilization of vegetation indices (VIs) derived from spectral band and different modeling techniques to detect early changes caused by the diseases (Singh et al., 2020; Marvasti-Zadeh et al., 2023). While RS applications for DNB have shown significant progress, the potential of these technologies for addressing BSNB remains largely untapped. Currently, there is a notable lack of direct studies or models developed specifically for L. acicola, limiting the ability to monitor and predict the disease’s impact. Addressing this gap by adapting RS techniques is important for enhancing detection, identifying spread pattern, and improving surveillance methodologies.

Traditional methods like on-site observations are essential but labor-intensive, time-consuming, and spatially constrained (Martinelli et al., 2015; Schnebele et al., 2015). In contrast, RS enables disease monitoring by assessing epidemic distribution, severity, and vegetation health through multi-scale and time-series analyses (Stone & Mohammed, 2017; Lausch et al., 2018; Panzavolta et al., 2021; Han et al., 2022; Lassalle & Fabre, 2022; Cao et al., 2022). While RS reduces the dependence on extensive field sampling, it complements ground-based observations to provide a more comprehensive understanding of disease dynamics (Yuan, Liu & Zhang, 2017; Pang et al., 2019; Zhang et al., 2019), overcoming the spatial and temporal limitations of discrete sampling sites.

Given the ability of L. acicola to cause significant economic and ecological damage, an important question arises: To what extent can remote sensing be utilized for the detection and monitoring of BSNB symptoms? To address this, we conducted a comprehensive bibliometric analysis using the Web of Science (WoS) database and used VOSviewer to visualize research trends based on published article keywords. The results showed a notable gap in the literature, with limited direct research on RS applications for BSNB detection and no prior bibliometric studies specifically focused on this disease. To broaden our understanding, we also examined RS applications in other pine needle diseases, such as Dothistroma Needle Blight. This analysis identified potential advantages and technological limitations to inform future approaches to BSNB detection.

This review addresses two key research questions: (1) whether RS and geospatial-based studies on BSNB have previously been published, and (2) what the advantages and limitations of RS applications in other pine needle disease research are, and whether these methodologies can provide a foundation for future BSNB investigations.

Survey methodology

This study follows a systematic bibliometric approach, consisting of three key stages: data collection, data processing, and visualization to assess research trends and gaps in BSNB detection using RS.

Data collection

A comprehensive literature search was conducted in the Web of Science database, applying well-defined search parameters (Table 1). The search was performed across all fields to ensure broad coverage of relevant literature. Boolean operators (AND, OR) were utilized to refine search results and ensure comprehensive retrieval of relevant studies.

Table 1 Search criteria and publication results.

Seach criteria	Purpose	Publications	
“Pine Needle diseases”	Recent publication trend for BSNB (Fig. 2)	653	
“Lecanosticta acicola” OR “Mycosphaerella dearnessi” OR “Scirrhia acicola” OR “Brown Spot Needle Blight” AND “detection method” AND “Remote Sensing”	Specific focus on BSNB detection using RS (Fig. 3)	84	
“Pine disease” AND “Remote Sensing”	General remote sensing applications in pine disease detection (Fig. 6)	173	

The dataset spans publications from 1929 to January 2025. The starting year of 1929 was selected because it marks the earliest availability of research data in the Web of Science database. The collected dataset was exported in RIS (Research Information Systems) format for further processing and analysis.

Data processing and visualization

Following data extraction, bibliometric analysis was conducted using VOSviewer (version 1.6.3) to identify key research trends, co-occurrence networks, and thematic clusters (Van Eck & Waltman, 2017). Bibliometric maps were generated to visualize collaborative relationships, research hotspots, and existing gaps (Zhao, Tang & Zou, 2019; Jia & Mustafa, 2022). A flowchart (Fig. 1) outlines the complete data collection and processing workflow.

Figure 1 Methodology flowchart of VOSviewer software.

Interpretation of bibliometric maps

The final stage involved analyzing thematic clusters and research landscapes. The network visualizations generated by VOSviewer highlighted interconnected research domains. This facilitated the classification of key research trends, knowledge gaps, and opportunities for future exploration.

Results

Bibliometric analysis (research trends)

Figure 2 shows various pine diseases, their causal pathogens, and associated key terms, reflecting research focus areas based on published studies over time. The color gradient (blue to yellow) represents publication years, with blue (~2012) for older studies, green (~2014–2016) for intermediate research, and yellow (~2018) for recent studies. Strongly interconnected nodes highlight well-established research areas while emerging topics indicate growing scientific interest. BSNB, though historically less studied, has recently gained attention, particularly concerning its relationship with climate change and host susceptibility.

Figure 2 Bibliometric network visualization of needle disease research trends.

Figure 3 presents a co-occurrence network of key terms in BSNB research, showing relationships among major clusters. Larger nodes indicate frequently occurring keywords, while different colors represent thematic clusters. The network highlights strong associations between Brown Spot Needle Blight, pathogen dynamics, climate change, and disease resistance, reflecting active research on host range, environmental influences, and molecular factors. The presence of terms related to genetic studies, dispersal mechanisms, and host-pathogen interactions suggests ongoing efforts in epidemiology. However, remote sensing applications are notably absent, supporting the need for further exploration in this domain.

Figure 3 Co-occurrence network of BSNB Research.

The following sections provide a detailed investigation into major aspects of BSNB research. This includes an analysis of BSNB’s global distribution, symptomatology, the impact of climate change on its spread, and existing disease detection methods.

Global distribution and quarantine status

Van Der Nest et al. (2019) provided a comprehensive review of the host and geographic range of L. acicola, identifying 53 pine species and hybrids affected by BSNB. The study confirmed its presence in 31 countries, with the majority of reports originating from North America and Europe, followed by Asia, while the disease unreported in Africa, Netherlands (confirmed by survey), Belgium (confirmed by survey), and Sweden (pest eradicated) (EPPO, 2024b). In North America, L. acicola was first recorded in 1876 on P. echinata (de Thümen, 1878) and has since been found on various native and exotic pine species, including P. taeda, P. elliottii, P. ponderosa, P. palustris, and P. strobus (Skilling & Nicholls, 1974). In Europe, BSNB was first reported in 1942 on P. radiata in northern Spain and has since spread to 17 countries (Ortíz de Urbina et al., 2016; García-García et al., 2025).

Severe outbreaks have reported in Austria (P. mugo, P. sylvestris), Italy (P. mugo), Slovenia (P. mugo, P. sylvestris), and Switzerland (P. mugo, P. uncinate) (Brandstetter & Cech, 2003; Janoušek et al., 2016; Sadiković et al., 2019; Holdenrieder & Sieber, 1995). In Asia, the pathogen has been documented in plantations of non-native species such as P. thunbergii, P. elliottii, and P. taeda in China, with severe damage reported (Huang, Smalley & Guries, 1995). Additional susceptible species include P. caribaea, P. palustris, and P. echinata, while native Chinese pines like P. taiwanensis show resistance (Huang, Smalley & Guries, 1995; Chuandao et al., 1986). The disease has also been found in Japan and South Korea, though infections in the latter remain mild (Suto & Ougi, 1998; Seo et al., 2012). Due to its severe impacts, it has been classified as a quarantine pest, holding A1 status in regions such as Africa, Argentina, Chile, and Russia, and A2 status in Europe (EPPO, 2024a). The rising global incidence highlights the need for improved quarantine protocols and early detection methods.

BSNB symptomology

BSNB symptoms vary depending on the host species. They generally begin as small yellow, grey-green, or reddish-brown spots with distinct margins, which later darken to brown and may be surrounded by a yellow halo (Skilling & Nicholls, 1974) (Fig. 4). The presence of these characteristic brown spots on pine needles led (Siggers, 1932) to coin the term ‘brown spot needle blight.’ In some cases, these spots may appear resin-soaked, depending on the host species. Infected needles generally die from the tip downward and eventually fall, with the disease spreading upward from the lower branches (Skilling & Nicholls, 1974; Van Der Nest et al., 2019). As the infection progresses, lesions enlarge, leading to tissue death and premature defoliation (Adamson, Drenkhan & Hanso, 2015). In North America, BSNB infections can occur throughout the year, but L. acicola produces the most spores during the summer months, peaking in June and August (Kais, 1975).

Figure 4 Close-up of infected needles and branch showing characteristic brown spots, some with yellow halos, and necrosis.

BSNB symptoms can sometimes be confused with Dothistroma Needle Blight since both diseases cause needle discoloration, but BSNB can be positively identified with microscopy or molecular analysis. One key morphological difference is that DNB typically forms distinct red bands around the infection site, while BSNB produces brown spots. In some DNB cases, the red bands in DNB can appear dark enough to resemble brown spots, which can lead to misdiagnosis (Barnes et al., 2016). Morphological identification of BSNB-causing fungus can be used to identify these needle characteristics as well as the morphological fungal characteristics of cultured samples (Dreaden et al., 2024). Molecular identification can also provide diagnostic insights, either using conventional PCR that target known diagnostic genes (Groenewald et al., 2007; Ioos et al., 2010) or by using Loop-Mediated Isothermal Amplification (LAMP) assays, a rapid and efficient alternative to traditional PCR (Aglietti et al., 2021).

Climate-driven disease dynamics

Variations in temperature and water availability play a significant role in shaping forest health and the distribution of native pathogens (Ramsfield et al., 2016; Kännaste et al., 2023). As with other fungi, the life cycle of L. acicola is strongly influenced by climatic conditions, which regulate key processes such as spore germination, mycelial growth, stomatal penetration, fructification body formation, and spore dispersal (Agan et al., 2021; Raitelaitytė et al., 2023; García-García et al., 2025). Laboratory studies indicate that conidial germination halts at temperatures below 5 °C during 72-h assays and at 32 °C after 48 h for northern lineage strains, whereas up to 80% of spores from southern lineage strains remain viable at this temperature (Siggers, 1944). Infection does not occur under controlled temperature cycles of 35 °C during the day and 27 °C at night, with a sharp decline observed after only 2 h at 38 °C. Field observations confirm that L. acicola infects needles in regions where the warmest month’s maximum temperature reaches 34.9 °C and the coldest month’s minimum temperature falls to −24.1 °C, with occurrences in areas experiencing as much as 543 mm of rainfall in the wettest months and as little as 6 mm in the driest months (Tubby et al., 2023). The increasing frequency of L. acicola outbreaks has been closely linked to recent climatic shifts (Broders et al., 2015; Wyka et al., 2017), suggesting that future climate trends may further influence its spread and severity (Ogris et al., 2023).

Recent efforts have focused on improving predictive models to assess L. acicola distribution and its response to climatic variation. Ogris et al. (2023) applied species distribution modeling (SDM) to estimate the pathogen’s potential range expansion, showing that temperate and boreal forests are increasingly suitable for its establishment. Such distribution maps provide forest managers with a basis for evaluating risks associated with L. acicola and planning mitigation measures. Tubby et al. (2023) reported the growing presence of L. acicola in European forests, highlighting its ability to persist across a range of climatic conditions. García-García et al. (2025) developed a model for predicting spore abundance in Atlantic climate regions, identifying temperature and precipitation as key factors influencing sporulation and dispersal.

Additional studies have examined spore dispersal patterns in greater detail. Wyka et al. (2018) analyzed the effects of temperature and precipitation on L. acicola spore dispersal and defoliation in P. strobus, comparing litterfall caused by defoliation with that from natural needle abscission. Mesanza et al. (2021) investigated weather conditions affecting spore release in northern Spain, further clarifying the role of climatic factors in pathogen spread. Mullett et al. (2018) examined L. acicola spore dispersal in P. radiata, showing its dependence on climatic conditions and the need for improved strategies. These studies highlight the importance of combining predictive modeling, climate data, and disease monitoring to track and manage L. acicola. As environmental conditions continue to change, early detection and targeted control measures will be essential in minimizing their impact.

Pandit et al. (2020) examined the influence of tree characteristics and climatic variables on foliar disease outbreaks in southern pines (P. taeda, P. palustris, and P. elliottii). Their study focused on (i) the relationship between foliar disease occurrence and tree-level traits, including crown ratio and diameter, and (ii) the impact of climatic variables, specifically mean dew point temperature, maximum vapor pressure deficit, and cold-season precipitation, on disease incidence at the landscape level. Using the MaxEnt model, a widely applied species distribution modeling (SDM) approach, they analyzed presence-only data to assess disease distribution. With Forest Inventory and Analysis (FIA) data, they investigated needle cast diseases of southern pines, including pathogens such as Lecanosticta acicola, Lophodermium Chevall., Lophodermella Hohn., Ploioderma Darker., and Hypodermia. The study identified crown ratio as a key factor influencing disease occurrence (p < 0.1) and found that climatic variables, particularly mean dew point temperature, maximum vapor pressure deficit, and cold-season precipitation, significantly affected disease incidence. The MaxEnt-based SDMs further highlighted the role of climate in shaping foliar disease patterns, reinforcing the importance of climate-based monitoring and management strategies for pine forest health.

BSNB detection methods

Traditional disease detection relies on visual symptom assessment, which, while cost-effective, lacks the specificity and sensitivity of molecular diagnostics (Baldi & La Porta, 2020). Molecular methods enable precise pathogen identification, even in asymptomatic plants (Luchi, Ioos & Santini, 2020). Standard techniques include polymerase chain reaction (PCR) and quantitative PCR (qPCR), which amplify pathogen-specific DNA sequences for accurate detection (Ioos et al., 2010). Loop-mediated isothermal amplification (LAMP) has emerged as a rapid, cost-effective alternative suitable for field applications without requiring advanced laboratory infrastructure (Aglietti et al., 2021; Patel et al., 2022). Unlike PCR, LAMP operates at a constant temperature, making it well-adapted for in-field monitoring (Patel et al., 2022).

Molecular genetic approaches also contribute to understanding the epidemiology and population structure of L. acicola (Marcet-Houben et al., 2024). Janoušek et al. (2016) applied microsatellite markers and Bayesian computation to investigate its origin, genetic diversity, reproductive strategy, and spread. The first genome-wide study of L. acicola assembled a reference genome and analyzed 70 natural isolates from northern Spain. Most belonged to the southern lineage but showed signs of introgression with northern lineage isolates, indicating active mating between the two lineages (Marcet-Houben et al., 2024). This genetic exchange may influence the pathogen’s adaptability, geographic expansion, and long-term population structure. Phenotypic analysis based on enzyme activity profiling identified functional differences between the two lineages, with introgressed strains exhibiting enzyme activity patterns more similar to the southern lineage. These findings highlight the role of genetic admixture in shaping L. acicola populations and emphasize the need for ongoing monitoring to assess its impact on forest health.

Despite their accuracy, molecular techniques depend on field sampling, which may delay diagnosis (Yang et al., 2021). Remote sensing offers a scalable solution for disease surveillance by enabling spatial mapping of infections and reducing reliance on localized sampling. While it does not directly identify pathogens, it complements field observations, enhancing disease monitoring across landscapes (Stone & Mohammed, 2017; Han et al., 2022). Multi-scale and time-series analyses further support its application in tracking vegetation changes and disease progression (Lausch et al., 2018; Han et al., 2022).

Based on the limited direct research available on BSNB detection using RS, a study conducted by Adedapo et al. (2024) utilized a multispectral unmanned aerial vehicle (UAV) (Phantom 4) and a deep learning (DL) framework incorporating Single Shot Detector (SSD) and RetinaNet algorithms to detect both diseased and dead trees within the Knight property. The study highlights the effectiveness of DL algorithms in detecting pine disease outbreaks, providing data on the number and precise locations of diseased and dead trees to support forest management in the Southeastern United States. Apart from this study, no direct research has been conducted on BSNB detection. However, geospatial technology has been widely applied in climate modeling, as discussed in the above section.

While research on BSNB detection remains limited, RS has been applied to assess other forest health issues. The following sections discuss recent advancements in remote sensing technology, as well as machine learning and deep learning approaches, and their role in detecting various pine needle diseases.

Advancements in remote sensing for disease detection

Reflectance spectroscopy is the predominant technique for in-field vegetation analysis and has been widely applied in scientific research for ecological and environmental assessments (Pandey et al., 2017; Lechner, Foody & Boyd, 2020; Singh, 2024). This method is based on the interaction between plants and light, where radiation is either reflected, absorbed, or transmitted, depending on internal and external plant dynamics such as structure, chemical composition, and water content (Zahir et al., 2024). Pathogens disrupt these dynamics by altering plant pigmentation, water content, and tissue function, which leads to structural modifications that manifest as disease symptoms (Tardieu et al., 2017). The physiological and phenological changes in plants are intricately linked to the nature of the pathogen (Dixon, 2012). These alterations produce distinct spectral patterns that facilitate spectral discrimination to differentiate between healthy and diseased leaves and canopies (Golhani et al., 2018). Therefore, remote sensing serves as a “radiodiagnosis” approach for plant disease investigation, allowing continuous monitoring over broad spatial scales (Decuyper et al., 2022; Singh, 2022a, 2022b).

RS technology involves the use of different platforms and sensors (Fig. 5). Spaceborne remote sensing enables large-scale disease detection and monitoring by capturing extensive spatial data (Massey et al., 2023). Airborne platforms, equipped with advanced RS capabilities, facilitate forest monitoring at finer spatial resolutions. These systems offer operational flexibility and the ability to generate high-resolution imagery. UAVs, in particular, provide low-altitude flight capabilities and very high-resolution imaging, making them particularly effective for the early detection of disease outbreaks (Berie & Burud, 2018; Singh, 2022a). Their adaptability enhances responsiveness in disease monitoring and helps overcome limitations associated with satellite-based observations (Li et al., 2020).

Figure 5 Remote sensing platforms and sensor combinations.

The fundamental RS sensor is an optical imaging system, similar to a conventional camera, capable of collecting data beyond the visible spectrum, extending into the infrared and thermal regions (Reddy, 2018). These sensors come in various configurations: multispectral sensors capture a limited number of broad bands, whereas hyperspectral sensors acquire thousands of narrow spectral bands, providing a more detailed spectral profile. Radar-based RS, an active system operating in the microwave region, is particularly effective under cloudy conditions due to minimal atmospheric absorption. Both lidar (Light Detection and Ranging) and SAR (Synthetic Aperture Radar) are active RS technologies that emit energy pulses and record reflected signals. Lidar operates within the visible to near-infrared spectrum and, as an active sensor, functions independently of external light sources, making it suitable for nighttime data acquisition (Lohar et al., 2021). SAR, in contrast, possesses unique capabilities, such as detecting surface roughness and land cover characteristics (Ouchi, 2013). Unlike lidar, SAR can penetrate dense cloud cover and operate effectively in darkness, whereas lidar provides high-resolution spatial data under clear sky conditions or at night. These active sensors provide distinct advantages for data collection under various environmental conditions. The utilization of multiple remote sensing platforms and sensors holds the potential to detect and monitor various forest diseases.

Integration of machine learning and deep learning approaches

Machine learning (ML) techniques are increasingly used for detecting and monitoring fungal diseases in vegetation (Jackulin & Murugavalli, 2022; Goyal, Verma & Kumar, 2025). These algorithms analyze patterns in large datasets, detecting disease progression and identifying high-risk forest areas (Selvaraj et al., 2020; Chehreh, Moutinho & Viegas, 2023). To improve accuracy, ML models integrate data from multiple sources, including climate, soil, and forest inventory datasets (Zhou et al., 2025). This approach enhances the understanding of environmental factors influencing disease outbreaks (Shivaprakash et al., 2022) and supports informed decision-making for forest management (Nitoslawski et al., 2021; Gavilanes Montoya, Castillo Vizuete & Marcu, 2023). ML methods are classified into supervised and unsupervised learning (Annabel, Annapoorani & Deepalakshmi, 2019). Supervised learning uses labeled datasets to predict outcomes, while unsupervised learning identifies patterns in unlabeled data (Zaadnoordijk, Besold & Cusack, 2022).

Deep learning (DL), a subset of ML, uses neural networks to extract patterns autonomously and is effective in image recognition and disease classification (França et al., 2021; Jung et al., 2023; Upadhyay et al., 2025). Several ML and DL techniques, including convolutional neural networks (CNNs) (Lu, Tan & Jiang, 2021), K-nearest neighbor (KNN) (Goel & Nagpal, 2023), artificial neural networks (ANNs) (Golhani et al., 2018), and support vector machines (SVMs) (Annabel, Annapoorani & Deepalakshmi, 2019), have shown high accuracy in plant disease detection. Despite advancements, RS-based ML/DL applications for BSNB detection remain largely unexplored. Further research could improve early detection and disease monitoring.

Assessing disease symptoms using multi-source remote sensing: needle disease case studies

For the second objective of this article, which explores RS applications in pine needle disease research, this section reviews key studies. RS has been extensively utilized for disease detection and progression monitoring in pine forests (Fig. 6). Techniques such as hyperspectral imaging, multispectral, lidar, and satellite-based observations have played a key role in identifying early symptoms, tracking disease dynamics, and developing predictive models. These approaches enable large-scale disease surveillance and support proactive management strategies.

Figure 6 Bibliometric network visualization of research trends related to pine needle diseases and remote sensing.

The network illustrates major themes, including disease classification, climate change, spectral analysis, and ML/DL-based detection. While Pine wilt disease (PWD) is a vascular wilt disease and not a needle disease, its frequent association with RS studies suggests its role as a leading case study in ML/DL applications for pine disease detection.

RS-based research on pine needle diseases has evolved significantly, with early studies laying the foundation for disease assessment using airborne and field-based spectral analysis. Coops et al. (2003) used airborne hyperspectral imagery (CASI-2) to assess Dothistroma Needle Blight in P. radiata plantations in Australia, finding strong correlations between crown reflectance and ground-based severity estimates. This study laid the foundation for canopy health assessment using hyperspectral remote sensing. In the same year, Stone, Chisholm & McDonald (2003) employed portable chlorophyll fluorometry to analyze DNB-affected P. radiata needles, identifying the 709/691 nm reflectance ratio as a key spectral indicator of disease severity.

Beyond hyperspectral approaches, thermal and lidar-based techniques have also been used to detect pine needle diseases. Smigaj et al. (2019a) used UAV-borne thermal imaging and lidar to detect Red Band Needle Blight (RBNB) in P. sylvestris, finding significant correlations between crown temperature and disease severity (R2 = 0.27–0.41). In a related study, Smigaj et al. (2019b) combined hyperspectral and lidar data to classify infected trees, achieving 80.9% accuracy, which improved to 96.7% using stepwise discriminant function analysis. Expanding to large-scale monitoring, Watt et al. (2021) analyzed 6,276 observations of DNB severity in New Zealand’s P. radiata plantations over 37 years, comparing parametric and non-parametric models to develop a fine-scale disease severity map. Watt et al. (2023) further refined this approach using UAV-collected hyperspectral data, integrating 3D radiative transfer models (PRO4SAIL) with Random Forest algorithms, improving prediction accuracy from R2 = 0.52 to R2 = 0.85.

As Sentinel-2 satellite imagery has proven effective for forest disease monitoring, its 13 spectral bands, particularly the red-edge band (705–740 nm), have been widely used to detect chlorophyll content changes, water stress, and early symptoms of needle diseases (Wang et al., 2018; Zhang et al., 2018). Meneghini et al. (2022) demonstrated its utility for detecting White Pine Needle Damage (WPND) in P. strobus forests, identifying NDII (Normalized Difference Infrared Index) as the most effective index for tracking defoliation, with early July as the optimal detection window. While severe WPND was detectable, light to moderate damage remained challenging, with classification accuracies not exceeding 75%. The study emphasized the need for higher spatial resolution sensors (e.g., PlanetScope, WorldView) and plot-based biochemical sampling to improve detection accuracy.

More recently, Timalsina et al. (2024) explored hyperspectral RS and foliar traits for WPND detection. Their study demonstrated that a Random Forest (RF) model based solely on spectral vegetation indices (SVIs) achieved 87% accuracy and a Kappa coefficient of 0.68, effectively classifying trees as asymptomatic or symptomatic. When field-measured foliar traits were combined with RS data, accuracy dropped to 77%, with a lower Kappa coefficient (0.46). These findings highlight the potential of hyperspectral data and machine learning models for timely WPND detection, aiding forest management. The study suggests that higher-resolution hyperspectral data from UAVs could improve local-scale assessments, while Sentinel-2’s red-edge bands could enable broader landscape-level WPND monitoring. Building on these advancements, Watt et al. (2024) developed a novel methodology integrating Sentinel-2 satellite imagery and climatic data to predict Red Needle Cast (RNC) outbreaks in P. radiata forests in New Zealand 7–8 months before peak disease expression. The study utilized the Red/Green Index Difference (R/G) to classify areas as disease-free or showing RNC symptoms, with solar radiation, relative humidity, rainfall, and maximum air temperature identified as key climatic predictors. Using 1,976 plots, a RF model was developed, achieving 89% accuracy and an F1 score of 0.89 in predicting RNC incidence. This approach enables large-scale, early disease detection, allowing for targeted monitoring and treatment to mitigate the disease’s impact before its peak expression in spring.

While Pine Wilt Disease (PWD) is a vascular wilt disease rather than a needle disease, it has been extensively studied in remote sensing-based detection using ML and DL models (Fig. 6). Its frequent inclusion in RS applications highlights its value as a reference system for automated disease monitoring in pine forests. The well-established methodologies used for PWD detection provide a foundation for developing similar approaches tailored to pine needle diseases. However, its inclusion in this study is based solely on its relevance to RS-based methodologies, rather than any direct biological comparability to BSNB or other needle diseases.

PWD, caused by the pinewood nematode (Bursaphelenchus xylophilus), has been widely monitored using a combination of UAV-based multispectral imaging, high-resolution satellite data, and ML/DL-driven classification models. Mantas et al. (2022) demonstrated that Sentinel-2 imagery could detect PWD with 95% accuracy, while Yu et al. (2021a) applied UAV-based multispectral imaging, achieving classification accuracies between 60.98–66.7%. The integration of ML/DL models has significantly enhanced PWD detection capabilities, improving accuracy, efficiency, and scalability. Li et al. (2021) demonstrated the effectiveness of CNN-based models such as YOLOv4-Tiny-3Layers, achieving 84.88% precision for airborne edge computing. UAV-based Faster R-CNN models further improved classification accuracy, reaching 60.98–66.7% (Wu et al., 2021), while Zhang et al. (2022b) achieved 95.24% accuracy using SVM and Genetic Algorithm models on hyperspectral UAV data, showcasing the potential of advanced classification techniques. Beijing-2 satellite data combined with CNN models achieved 99.4% accuracy in rapid tree identification (Zhou et al., 2022b), further demonstrating the potential for scalable disease monitoring. The application of deep learning methods continues to enhance PWD detection, with models such as YOLOv5-PWD integrating UAV and satellite imagery for large-scale surveillance (Cai et al., 2023). Additionally, hybrid approaches leveraging UAV-based hyperspectral data and ML models, such as RF and SVM, have shown high accuracy, with some studies achieving over 95% precision in monitoring PWD infection stages (Zhang et al., 2022b). The growing reliance on ML/DL for PWD detection highlights its crucial role in precision forestry, offering cost-effective, scalable, and highly accurate solutions for disease monitoring. Table 2 provides a comprehensive summary of ML/DL models applied to PWD detection, highlighting the various methodologies, accuracies, and research gaps.

Table 2 Different machine learning and deep learning methods for pine wild disease detection.

Host	Data	Research
contribution	Accuracy	Research gap	Reference	
Pine	Quick Bird, UAV; Bayesian Network	Effective modeling of PWD	QB: UA: 89.19%, PA: 86.84%; UAV: UA: 90%, PA: 92.31%	Lack of effective tools for modeling PWD	Huang et al. (2013)	
Pine	UAV (RGB); Faster-RCNN, RPN	Detection of dead trees	DA: 90%	Optimization needed	Deng et al. (2020)	
Coniferous	UAV (Multi- & Hyperspectral); RF	Feasibility of RF for PWD detection	CA: 0.91	Early detection exploration	Iordache et al. (2020a)	
Pine	Airborne (Multi- & Hyperspectral); RF	ML algorithms for PWD detection	0.91	Early detection methods	Iordache et al. (2020b)	
Pine	Airborne Hyperspectral, Lidar; RF	PWD infection stage detection	HI: 66.86%, Kappa: 0.57; LiDAR: 45.56%, Kappa: 0.27	Utilization of Lidar and HI data	Yu et al. (2021a)	
Coniferous	UAV (RGB); YOLOv4-Tiny-3Layers (DL)	Airborne edge-computing and Lightweight DL-based system	AP: 84.88%	Effective detection with DL	Li et al. (2021)	
Coniferous	UAV (Multispectral); CNN (DL)	Object detection (dead & brown trees) with multichannel CNN	mAP: 86.63%	Multichannel CNN implementation	Park et al. (2021)	
Mixed Forest	UAV (Multispectral); Faster R-CNN, YOLOv4 (DL)	Systematic infection stage division	R-CNN: 60.98–66.7%, YOLOv4: 57.07–63.55%	Stage identification with UAV and DL	Yu et al. (2021b)	
Pine	UAV (Multispectral); Faster R-CNN, YOLOv3	Cost-effective early diagnosis	Precision: 0.60–0.64	Large-scale rapid screening	Wu et al. (2021)	
Coniferous	UAV (RGB); Various ML algorithms	PWD detection method	ANN: 0.99
(best model)	Cost-effective UAV imagery with ML	Oide, Nagasaka & Tanaka (2022)	
Pine	UAV (HI strips); 3D convolutional layers, transformer blocks	Fine pixel-level detection	F1-score: 0.9	Improved model generalization	Li et al. (2022a)	
Pine	Beijing-2; RF	Online monitoring with spatial-spectral features	86.66%	Efficiency with satellite data	Zhang et al. (2022a)	
Pine	UAV, Landsat 8; MA-UNet (DL)	Improved pest detection	Recall: 57.38%	Addressing detection challenges	Ye et al. (2022)	
Pine	UAV (Multispectral); DDYOLOv5, ResNet50 (DL)	Improved detection, severity classification	Precision: +13.55%, Recall: +5.06%, F1: +9.71%	Accuracy and classification enhancement	Hu et al. (2022)	
Pine	UAV (Hyperspectral); SVM, GA	Monitoring and estimation	Accuracy: 95.24%, Kappa: 0.9234	Efficient monitoring methods	Zhang et al. (2022b)	
Pine	Geospatial Data; CA-Markov Model	Prediction and analysis of PWD occurrence trends	CA-Markov: 93.19%, Grid: 95.19%, Kappa: 0.65	Spread prediction on geospatial scale	Liu & Zhang (2022)	
Pine	UAV (Visible, Multispectral); MFTD (DL)	Early-stage detection	Precision: 0.90	Multi-band image-fusion development	Zhou et al. (2022a)	
Pine	UAV (RGB); Unsupervised method	Adaptive threshold, spatial clustering	F1: 91.35% and 0.8373	Rapid monitoring with decision fusion	Wan et al. (2022)	
Pine	Sentinel 2, UAV (RGB); RF	Detection using stochastic model	R2: 0.88	Effective method with physical model	Li et al. (2022b)	
Pine	Sentinel 2; ML	Tree decline detection	95%	Decline detection algorithm	Mantas et al. (2022)	
Pine	Airborne imagery; MSSCN, Gaussian kernel	Detection of standing dead trees after PWD outbreak	Precision: 0.94, Recall: 0.84, F1: 0.89	Multi-scale spatial information use	Han et al. (2022)	
Pine	Beijing 2; CNN and bounding box tool	Rapid identification and location of infected trees	99.4%	Quick tree identification	Zhou et al. (2022b)	
Pine	UAV (Multispectral); RF, SVM, LDA	Monitoring windows for infection stages	RF: 0.61	Optimal monitoring periods (early, middle, and late stages)	Wu et al. (2023)	
Pine	UAV (RGB), Sentinel-2; YOLOv5-PWD (DL)	Drone and satellite imagery framework	1.9%	Detection accuracy improvement	Cai et al. (2023)	
Pine	UAV (Multispectral); CNN (PWDNet), Balance Mixup	PWD prevention and control	Precision: 0.90	High recall and precision methods	Rao et al. (2023)	
Note:

PWD, Pine Wilt Disease; UAV, Unmanned Aerial Vehicle; DL, Deep Learning; ML, Machine Learning; RF, Random Forest; CNN, Convolutional Neural Network; RPN, Region Proposal Network; HI, Hyperspectral Imaging; UA, User Accuracy; PA, Producer Accuracy; DA, Detection Accuracy; CA, Combined Accuracy; AP, Average Precision; MAP, Mean Average Precision; ANN, Artificial Neural Network; GA, Genetic Algorithm; CA, Cellular Automata; MFTD, Multi-Band Image-Fusion Infected Pine Tree Detector; MSSCN, Multi-Scale Spatial Supervision Convolutional Network; SVM, Support Vector Machine; LDA, Linear Discriminant Analysis; R2, Coefficient Of Determination; Kappa, Kappa Statistic.

Challenges and future outlook

Remote sensing has significantly improved forest disease detection and monitoring, yet several challenges and limitations persist. A primary issue is distinguishing biotic and abiotic stressors due to overlapping spectral characteristics. Multispectral and hyperspectral imaging allow vegetation index calculations, such as normalized difference vegetation index (NDVI), linked to chlorophyll degradation and early stress responses. Thermal sensing detects canopy temperature anomalies from altered transpiration, serving as a potential early indicator of infection. Lidar enables individual tree assessments by detecting defoliation patterns and canopy thinning, essential for tracking disease progression. Integrating these sensors through multi-source data fusion holds strong potential for improving BSNB detection accuracy at both stand and landscape scales.

While correlation analyses and spectral indices aid differentiation, biochemical validation remains essential for improving RS-based classification accuracy (Girma et al., 2005; Fahey et al., 2020). Ground truthing ensures remote sensing data reliability by confirming stressors through field studies and laboratory analyses. Integrating remote sensing with ground truthing improves disease identification and quantifies forest areas affected by BSNB, other needle pathogens (needle cast), and abiotic stressors. A combined RS-based and field approach strengthens BSNB detection and improves understanding of its ecological impact. Despite similarities to Dothistroma Needle Blight, BSNB remains underexplored in RS-based studies. Hyperspectral imaging effectively detects foliar diseases like DNB (Coops et al., 2003) but requires validation to differentiate BSNB from abiotic stressors and co-occurring pathogens. A major limitation in RS-based disease monitoring is the inherent trade-off between spatial and temporal resolution. While high-resolution data improve detection accuracy, operational implementation is often constrained by acquisition costs and revisit frequency, limiting feasibility for large-scale applications (Samadzadegan, Toosi & Dadrass Javan, 2024).

Although Pine Wilt Disease is distinct from needle diseases, remote sensing techniques, including UAV-based multispectral imaging and high-resolution satellite data, have been effectively used for its detection. ML/DL-driven classification models have demonstrated high accuracy in identifying symptomatic trees at various infection stages, supporting early disease management. While PWD affects the vascular system, needle diseases primarily impact foliage, progressing gradually. However, advancements in remote sensing methodologies developed for PWD can be adapted to improve BSNB detection, provided that the differences in disease progression and symptom presentation are carefully considered. ML/DL models trained on spectral signatures and vegetation indices can enhance early symptom identification, while multi-sensor data fusion can strengthen large-scale BSNB monitoring.

Geospatial climate models play a key role in assessing BSNB risk. Temperature, humidity, and precipitation influence disease progression by affecting pathogen sporulation, dispersal, and host susceptibility. Integrating RS-derived disease indicators with climate models can improve risk assessments and support the development of early warning systems. While RS has been applied to monitor various needle diseases, further research is needed to fully integrate it with climate modeling for BSNB risk prediction. Combining UAV and satellite-based RS with climate modeling can provide a more comprehensive understanding of BSNB dynamics. Developing predictive models and risk maps that incorporate both RS and climatic data would enhance early detection and disease management strategies.

Future advancements require optimizing multi-sensor data fusion, improving ML/DL adaptability, and strengthening climate-RS integration. High-temporal-resolution remote sensing data can bridge monitoring gaps, while UAV-based high-resolution imaging remains essential for localized assessments. Collaboration between remote sensing experts, forest pathologists, and climate scientists will refine predictive models and improve large-scale disease monitoring. Despite challenges, continued advancements in sensor technology, data analytics, and computational modeling will support scalable, proactive BSNB detection and management strategies.

Additional Information and Declarations

Competing Interests

The authors declare that they have no competing interests.

Author Contributions

Swati Singh conceived and designed the experiments, performed the experiments, analyzed the data, prepared figures and/or tables, authored or reviewed drafts of the article, and approved the final draft.

Lana L. Narine conceived and designed the experiments, analyzed the data, authored or reviewed drafts of the article, supervision, Funding Acquisition, and approved the final draft.

Janna R. Willoughby analyzed the data, authored or reviewed drafts of the article, and approved the final draft.

Lori G. Eckhardt analyzed the data, authored or reviewed drafts of the article, Funding Acquisition, and approved the final draft.

Data Availability

The following information was supplied regarding data availability:

This is a literature review.

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
