# Peer review of "Remote sensing-based detection of brown spot needle blight: a comprehensive review, and future directions"

_PeerJ, doi:10.7717/peerj.19407_

## Round 0.1 · original submission · Major Revisions

Dear authors,

I ask you to very carefully improve the manuscript in accordance with the recommendations of the anonymous reviewers. The direction of your research is quite new, interdisciplinary. Therefore, I hope that this article will arouse considerable interest among readers. The greater is the responsibility of the authors: to present the data in a form that will become standard for many years to come for this direction of research. I hope that eliminating the shortcomings indicated in the very detailed review of the 2nd reviewer will help you successfully pass the second review and publish an improved version of this manuscript.

Reviewer 1 ·

Basic reporting

The literature reviewed might have been analyzed more or critiqued. This might have explored the gaps in knowledge and lead to new research questions. The gaps are explored but it would have been better if that is also made with more examples in forest disease management highlighting their pros and cons. The language need some revision and some are highlighted

Experimental design

Need to analyze the literature with more depth.

Validity of the findings

It is commendable that the authors have highlighted future directions. However, instead of merely mentioning them, a more detailed roadmap would strengthen the manuscript. This roadmap could outline how multifaceted approaches can be utilized, with a discussion of their respective strengths and weaknesses. While some ML and DL approaches are referenced and elaborated upon in the supplementary files, these could have been further expanded in the main text to enhance the manuscript's depth and accessibility.

Annotated reviews are not available for download in order to protect the identity of reviewers who chose to remain anonymous.

Reviewer 2 ·

Basic reporting

The introduction section of the manuscript lays out background information regarding forest disturbances, the fungal causal agent of Brown Spot Needle Blight (BSNB) of pines, Lecanosticta acicola and its relevance as forest pathogen, disease detection methodologies, the use of remote sensing in forestry, the economic importance of pine forests in the southeastern United States and the insect pests that are known to threaten southeastern forests more broadly. The introduction builds to hypothesize that remote sensing may be a useful methodology to study BSNB, indicating that there is a lack of remote sensing research dedicated to detecting and monitoring BSNB.
While the introduction and background do address many of the key points of the hypothesis, the review is incomplete, inappropriate in focus, and fails to report key findings and developments in the study of L. acicola as both an invasive threat to pine forests worldwide and an emerging native disease of economically important timber pines in the southeast United States. The highlights and issues with the introduction are outlined in this section in brief, and in detail separately in a later section of this review. The introduction provides a superficial review of L. acicola and BSNB of pines, referring to some recent work but without summarizing the known host range, impacted regions, or recent findings regarding the nature of this pathogen as an emerging threat within its native range (as outlined by the extension articles: Datta et al 2021, Cunningham 2022, Meinecke and Eshleman et al 2024, the research article Meinecke et al 2024, and the research thesis Datta 2021) and as an invasive fungal disease outside North America (Tubby et al 2023, Marcet-Houben et al 2024). The introduction also presents the first of several instances where White Pine Needle Damage (WPND), a disease of Pinus strobus for which the causal agents are thought to be a group of cooccurring fungi (including L. acicola), is conflated with BSNB (a disease of other pines where L. acicola is the sole primary pathogen). These diseases bear dissimilar symptomatologies, etiologies, and implications, and are simply not identical conditions that can be referred to interchangeably.

The introduction presents a succinct and accurate review of the utility of remote sensing in forestry research and management. The point is made later in the manuscript that validated laboratory diagnostic assessments should support remote sensing for disease detection, and this point should also be emphasized in the introduction. However, the omission of any mention of research on the well-studied Dothistroma Needle Blight of pines, Swiss Needle Cast of Douglas fir, and Red Needle Cast of conifers is surprising, as these are prominent (and in many cases catastrophic) needle diseases of commercially important conifers worldwide for which remote sensing has been successfully employed to facilitate research and management. In my opinion, the lack of consideration of these systems and the use of remote sensing to study and manage them in the introduction (and manuscript more broadly) indicates an incomplete review.

One paragraph of the introduction is used to frame geographical context of the study, pine forests of the southeastern United States. The system is initially well represented, however a heavy emphasis is placed on insect pests of all major forest tree species in the southeastern US without any mention of fungal diseases in this region. This shift in focus is abrupt and seemingly off-topic. The authors later make the case that remote sensing enables pathogen-specific detection, so a focus on the threat of insect pests, and not emerging fungal diseases in the region (including pine needle diseases), is out of place and does not serve the review.

As written, I do not feel that the introduction is sufficiently complete to provide context and justification for the goals of the review. Many of the references used are a decade or more out of date, and the introduction should be revised with the most recent literature. This is especially important as the emergence of BSNB as a domestic and international threat is a recent phenomenon and a dynamic situation of which our understanding is rapidly developing. The introduction is not sufficiently complete to adequately present BSNB as a critical emerging forest health threat or to properly frame remote sensing as a promising tool for this pine needle disease.

I feel this review, when complete, will be an extremely important for the forest health community, both within the subject region of the southeastern United States and globally. Lecanosticta acicola is, with certainty, one of the most important emerging forest pathogens globally and the impact of BSNB bears grave potential implications to pine forestry. Insights into the application of remote sensing to facilitate research and management of this disease is needed by the global forest health community.

Regarding figures:

Figure 1 – Please check that correct capitalization is used for the utilized products and services.

Figure 2- This is an interesting figure and helps the reader understand the information synthesized in the review. As presented, some of the nodes do not have labels. Please provide labels for all nodes. The caption does not describe how the figure was created or what the edges represent, please provide this information. Lastly, please provide a legend for the colors and the cluster types that they represent, and a scale of the node sizes.

Figure 3 – The legend should be amended to reflect the specific species depicted, which appears to be Pinus taeda. Symptom presentation differs among the pines present in the southeast United States and specificity here is critical. Photos A and C are not representative of BSNB symptom presentation as currently shown. Photo A is low-resolution and the discoloration and necrosis are hard for me to differentiate from other parts of the photo, and I routinely work with BSNB. A different photo should be used to better highlight in situ symptoms for less familiar readers. Photo B is great. Additional detail of needle symptoms would be useful for readers. Photo C is also not representative of whole-tree BSNB symptoms, as the browning of the lower crown is not particularly apparent. I suggest replacing photo C with one or more photos that depict obvious discoloration and defoliation and/or a comparison of diseased and healthy whole trees.

Figure 4 – As stated above for figure 2, some of the nodes do not have labels. Please provide labels for all nodes. Please include how the figure was created or what the edges represent. Lastly, please provide a legend node colors and sizes. Several nodes seem out of place considering the parameters used in the searches (cercopidae, israelensis, and phytophthora-ramorum), and are somewhat confusing as they do not appear to relate to BSNB. Are these artifacts of the search methodology?

Figure 5 – This is a very nice figure that is concise and clearly conveys the information. Please include in the caption an expansion of the acronyms SAR and LiDAR, so that the figure is self-standing.

Figure 6 – Please provide legends for the node size, node color, and edge thickness. As presented it is not possible to assess synergistic or antagonistic interactions among nodes.

Experimental design

Overall, the manuscript covers some of the key elements of the research question, namely an excellent general review of remote sensing applications, but is not sufficiently comprehensive to be considered a well-balanced study. Generally the review is well organized. However, the inclusion of an additional analysis of the literature regarding pine wilt disease is mentioned only in the results, and is not justified or mentioned in any earlier sections. Also, a consistent structure between the sections of the literature review and the sections of the recommended future directions may improved flow.

As written, this review does not adequately synthesize the literature regarding the disease system in question and therefore does not provide sufficient justification for the review. First, the text does address the historic and current threat that L. acicola and BSNB pose to pine forestry, its invasivity, and expanding damage to planted pine forests globally. Particularly, the BSNB pathogen is now considered one of the most pressing biotic threats to plantation forestry worldwide and has recently emerged as a major disease of industrial P. taeda forests in the southeastern US. Recently published research on the outbreaks in the southeastern US, caused by BSNB and other needlecasts, is not fully represented in the introduction. Without establishing the risks posed by BSNB and the need for additional research and monitoring, it is hard to convey the need for this review to the reader.

There is little attention given to the prior utilization of remote sensing to study and monitor other needle disease outbreaks in conifer species. By covering the previous successes, failures, and gaps identified by such studies, the authors may identify more efficient uses of RS for BSNB research and better outline potential pitfalls.

The incomplete review of the BSNB pathogen’s biology and etiology, the current understanding of risks posed by L. acicola, and the lack of research on the application of RS in other conifer needle diseases indicate that the manuscript is not sufficiently complete to be considered rigorous or ready for publication.
A critical issue with this manuscript is the narrow scope of the searches and the exclusion of other needle diseases that have been thoroughly studied using remote sensing. Lecanosticta acicola is only recently an emerging forest health threat, and only one of many globally important conifer needle diseases. The initial search on BSNB serves to identify the dearth of RS research on the disease, but fails to capture the lessons learned from the application of such technologies to the study of other similar disease systems. The case studies section would present an opportunity to close this gap, however only WPND is the only needle disease reviewed in this portion of the manuscript. The other case studies focus on a variety of other biotic and abiotic sources of damage.
Additionally, I am concerned as a pathologist about the ability of RS methods to discern between different types of needle disease and specifically between the damage caused by different necrotrophic fungi in needles of the same host species. The case studies highlight several examples of RS differentiating beetle damage from disease or wilt disease from other stressors. However there is no report of the use of RS to differentiate co-occurring needle diseases or any mention of the importance of this capability in areas with multiple needle diseases (eg Europe with BSNB and DNB, Oceana with DNB and RNC, or the southeast US with BNSB and other needlecasts). This may be a major hurdle to the use of RS technology in this system and should be addressed.

Lastly, while it understandable to have some lag between the literature search and the submission, I do feel that the literature search is not sufficiently recent for a manuscript submitted at the end of 2024. The authors include publications only through August 2023, missing the most current understanding of the BSNB invasion in Europe or the emerging outbreaks in the southeastern US. Lecanosticta acicola outbreaks worldwide are relatively recent and dynamic phenomenon, and I believe that this review must be as recent as possible to be useful to the reader.

Validity of the findings

Without an analysis of the implementation of remote sensing for other recent and relevant conifer needle diseases, I do not feel the authors have sufficiently supported the goal of exploring the utility of remote sensing to study BSNB. The case studies provide only a patchwork of potential uses and outcomes without a clear indication of how RS might succeed or fail when applied to pine needle disease outbreaks, and the recommended future directions are not thoroughly developed.

Additional comments

I am using this section to provide line-by-line comments and recommendations for the authors.
L25: replace “functioning” with function
L25-27: This is not universally true, not all biotic and abiotic disturbances exacerbate damage caused by the other. Please provide citations and consider some speaking on some relevant examples. There are many good examples of such interactions from globally important conifer needle disease systems (Dothistroma Needle Blight, Swiss Needle Cast, Red Needle Cast, among others).
L30: L. acicola itself is not a disturbance, rather the disease that it causes (BSNB) may be considered one.
L31: Scirrhia acicola (Siggers 1939) Phytopathology 29:1076-1077
L32-36: While it is certainly important to document the known hosts, it is also important to mention that not all hosts in all locations are impacted equally. Additionally, to properly motivate this review, I feel it is important to establish that BSNB is a devastating pathogen of many important timber pines, especially considering that an emphasis is later placed on the southeast United States.
L37: This pathogen also causes the necrosis and discoloration of the needle tissues that it infects, which are related to premature needle shed but are separate symptoms
L39: L. acicola was the first described Lecanosticta species, however I do not know of any work that has demonstrated the relative ages of the L. acicola lineages relative to the other species.
L41: It is important to also mention the association of L. acicola with WPND, and that this is a separate disease of P. strobus with differing symptomatology and etiology.
L41-44: While the Pandit 2020 is relevant to the southeastern United States, Wyka 2018 discusses WPND of P. strobus in the northeastern US, and Laas 2022 investigates genetic diversity in Europe. More appropriate assessments of the recent impacts of BSNB may be found in van der Nest et al 2019, Tubby et al 2023 (Europe), Ogris et al 2023 (global), Meinecke et al 2024 (southeastern US) and Datta et al 2021 (southeastern US). Additionally, as written this section appears to only refer to disease on P. palustris as there is no mention of the rising levels of damage on P. taeda.
L50: P. echinata
L51: Please see Oswalt 2019 for more current data.
L55: Please give a source for this claim, perhaps a report title. Additionally, this is a large number, but without also speaking to the uncounted damage caused by native species (e.g. L. acicola) it confuses the focus and seems out of place.
L55-66: While often considered together as causes of biotic disturbances, insect pests and microbial pathogens present different risks, have different impacts, and interact with abiotic stressors differently. As such, a focus on the pathogens of southern pines would be more appropriate. The insect pests may be worth mentioning, but not without diseases in a manuscript that otherwise focuses on a single fungal disease.
L62: Additional citations are necessary (if the insect pests are kept). Lycorma deliculata was not found in the US until 2012 and the latest citation given is Dukes et al 2009.
L65: I assume you mean to say the role that spatial configuration and management dynamics play in modulating L .acicola spread and BSNB disease pressure are not well characterized. In other regards, the spatial and management aspects of pine forestry in the southeastern US are exceedingly well characterized. There are numerous inventories of private industrial, small private, and public lands. To say these dimensions are not well characterized is a misrepresentation of the most productive and optimized regional forestry sector in North America.
L69-71: These terms should be explained for the review.
L94-101: Currently the goals of the review are not well-defined. A specific statement of the hypothetical uses of remote sensing to facilitate research and management of BSNB, and scope (outbreaks in endemic range vs invasive range) would help to guide the structure and better develop the argument of the review.
L107-114: My initial feeling is that there has been little application of remote sensing in BSNB research, so I am not certain that limiting the literature search to only BSNB will yield much information or direction. Instead I believe that a recognition of this dearth of information (as an excellent justification for the review) followed the review and analysis of the application of RS to the study of other prominent conifer needle disease would be much more constructive and provide insightful discussion points and tractable paths forward.
L152: Do you mean to say “upward through the crown”?
L167-168: Please specify the region and hosts that are the focus of the studies cited here, these dynamics can and often do vary by location and host as you highlight above in L157-158
L171-172: As mentioned above, the disease observed in P. strobus is WPND, not BSNB.
From L171 on: This paragraph appears to set up a reflection on the use of remote sensing for needle disease but the given examples do not flow logically nor do they summarize a cohesive list of studies that support the goals of the review.
L173-177: Broders did not conclude that L. acicola was the primary pathogen, but rather that it was the most commonly recovered and identified of several native pathogens associated with WPND. See McIntyre et al 2020 for a concise description of the complex of four fungal needle pathogens that are associated with this disease. The individual roles of these pathogens have not been disentangled at this time.
190: The authors cite Wyka et al 2017 while referring to back to content from Tubby et al 2023.
197: The symptoms listed are not characteristic of BSNB but are instead common to many needle diseases. I would recommend emphasizing that visual assessment alone is not enough to diagnose cases of BSNB and that observations should be supported by laboratory analysis.
195-203: There is no mention of culture-based diagnosis of BSNB, which is still widely used in many laboratories. A recent USFS publication by Dreaden et al (2024) outlines best practices for sampling, symptom assessment, isolation, and morphological identification of L. acicola and is specific to cases of WPND and BSNB in North America.
206: In addition to the general statement that detection of presymptomatic infections is possible by molecular methods (Luchi et al 2020), Meinecke et al 2024 demonstrates presymptomatic detection of L. acicola from P. taeda needles by LAMP and environmental DNA sequencing.
226: I would go so far as to say that molecular diagnostics are a necessary component to ground truth RS data, especially in contexts like the southeastern US where multiple different needle diseases co-occur on the same hosts (Datta 2021, Meinecke et al 2024).
238: Please change fungus to fungi.
239: Please correct to “Hypoderma”
246-270: These are excellent studies to highlight. I feel the authors should consider whether these studies was able to detect the needle pathogens alone, or if damage was detected only as an aggregate signal. These nuances are important to consider, as potential hurdles or pitfalls or matters that can be managed, when assessing whether RS will be useful for detecting BSNB against a backdrop of other biotic and abiotic disturbances.
259-260: Please check proper italicization of organisms and naming authorities.
277-280: The text appears to imply that the RS can detect pathogen-specific changes in host physiology, can the authors provide evidence that RS is capable of distinguishing among the damage caused by different necrotrophic fungal needle pathogens or perhaps emphasize that this is an area to be explored?
312: Please provide some explanation in the methods and perhaps here as to how the case studies were selected
328-336: This additional analysis appears rather abruptly, and there is no mention of it in the methods. All analyses should be outlined in the methods. Additionally, as stated above, I do not feel that Pine Wilt Disease is the most relevant system to review here. From a pathological perspective, wilts and foliar diseases function very differently, with different tissues suffering different levels of damage across different time scales. Considering that there is remote sensing research on conifer foliar diseases that more similarly manifest, spread, and impact stands (e.g. Dothistroma needle blight), I feel that more relevant information and conclusions can be drawn from reviewing that body of literature.
385: Bursaphelenchus xylophilus is a nematode, not a beetle.
420: Lin et al 2019 refers to pine shoot beetle damage, not a fungal blight
421: The ratio in question is shoot dieback ratio (SDR), not gunshot damage ratio
481: Several paragraphs in this section provide additional information and a list of possible tools, but do not frame questions or potential research plans. I feel this section especially should be revisited to emphasize what future work is needed specific to the gaps in BSNB research and how it can be addressed.
488-489: Are there cases where laboratory diagnostic validation is no longer necessary to support RS detection?
506-509: This sentence appears to undermine many of the other claims made in the manuscript. How do the authors propose utilizing RS for BSNB detection if it is unable to differentiate between different pathogens?
531-540: This section is excellent, it also contradicts some earlier claims that ground truthing is not needed after validation. Is the contradiction due to a mistake with wording?
544: This is problematic in the case of BSNB in the southeastern US where L. acicola is native and endemic. Functionally L. acicola is universally present in the region’s pine forests at some level. I would suggest instead emphasizing how disease pressure might vary with environmental and host factors.
556: In areas where improved pines are deployed, risk mapping can and should take genetics into account (Lim-Hing et al 2024).
570: “disease detection by remote sensing”
579: Please check the formatting of references and citations

---

## Round 0.2 · Minor Revisions

Dear authors, I kindly ask you to make changes to this manuscript so that the reviewers can re-evaluate your manuscript.

Reviewer 3 ·

Basic reporting

• Clarity and Language:
The manuscript is written in professional and clear English. However, some sections contain long and complex sentences, which may hinder readability. Minor grammatical refinements would enhance clarity. Consider simplifying sentence structures where possible.
• Literature Context and References:
The review includes an extensive reference list with relevant sources. The background on remote sensing and Brown Spot Needle Blight (BSNB) is well-developed. However, certain sections, such as the discussion on machine learning (ML) and deep learning (DL) applications, could benefit from additional references to recent works. A few citations appear outdated and should be supplemented with more recent studies.
• Manuscript Structure and Formatting:
The manuscript adheres to the PeerJ journal's format, with a clear introduction, methodology, results, and discussion sections. However, some paragraphs are lengthy and could be broken down for better readability. The figures and tables appear appropriate but should be checked for consistency in formatting and labeling.

Experimental design

• Scope and Aims:
The study aligns well with the journal’s aims and scope. The research question is relevant, addressing a critical gap in the application of remote sensing for BSNB detection. The rationale for the study is well justified.
• Survey Methodology:
The bibliometric analysis methodology is adequately described. However:
o The search strategy and inclusion/exclusion criteria could be more explicitly defined. Were there specific keywords, timeframes, or journal filters applied?
o The Web of Science database is appropriate, but additional databases (e.g., Scopus, Google Scholar) might improve coverage.
o The methodology could benefit from a reproducibility statement—how can future researchers replicate the bibliometric analysis?
• Organization and Logical Flow:
o The manuscript is logically structured. However, the transition between sections (particularly from results to discussion) could be smoother.
o The figures and bibliometric maps are useful but should be cross-referenced more frequently in the text to aid reader comprehension.

Validity of the findings

• Findings and Interpretation:
The review effectively synthesizes existing literature but does not present novel experimental findings, which is expected for a review article. The conclusions drawn are supported by the cited research.
• Critical Gaps in Research:
o The authors highlight research gaps, particularly the limited direct applications of remote sensing for BSNB detection.
o However, the review could provide more specific recommendations for future research, such as:
 Potential machine learning models suitable for BSNB detection.
 Emerging satellite sensors or UAV technologies that could improve detection capabilities.
 How ground-based field validation can support remote sensing approaches.
• Limitations and Bias:
o The authors acknowledge limitations but should discuss potential biases in their literature selection process.
o Comparisons with other disease detection studies could strengthen the argument for why BSNB remains understudied in remote sensing.

Additional comments

• Strengths of the Manuscript:
o The manuscript provides a comprehensive review of remote sensing applications for forest disease monitoring.
o The bibliometric approach is a novel addition, offering insights into research trends.
o The discussion of machine learning and geospatial modeling adds depth to the study.
• Areas for Improvement:
o Improve clarity by breaking down long paragraphs and simplifying sentence structures.
o Expand the methodology section to ensure reproducibility.
o Provide specific actionable recommendations for future research directions.
o Discuss alternative remote sensing techniques that could be tested for BSNB.

Final Recommendation
The manuscript is a well-organized and informative literature review with high relevance to forest health monitoring. However, minor revisions are needed to:
1. Improve clarity and readability.
2. Enhance the methodological description.
3. Strengthen discussions on research gaps and future directions.
Once these concerns are addressed, the manuscript will be suitable for publication in PeerJ.

·

Basic reporting

An interesting and well-written review. The topic of the review is certainly relevant. Step-by-step presentation of the material with detailed discussion. I would like to point out the following to the authors. Disease damage to a plant usually leads to a decrease in transpiration fluxes and corresponding changes in the temperature of the aboveground part of the plant / canopy. Shouldn't the description of approaches to remote diagnosis of plant damage by BSNB/disease(s) emphasize this approach to diagnosis of forest health?
The list of references is sufficient. However, for the authors' consideration, shouldn't you mention 1-2 pioneering studies?

Experimental design

No comment

Validity of the findings

No comment

---

## Round 0.3 · Major Revisions

Dear Doctor, I ask you to carefully analyze the comments of the reviewers. The shortcomings pointed out by the reviewers are very significant. I hope that the new version of this article can be accepted for publication.

Reviewer 2 ·

Basic reporting

I commend the authors for their hard work and substantial improvements from the first draft. I feel this is a valuable contribution to our efforts to understand and control BSNB in the southeast United States. There are some outstanding issues regarding the presentation of some forest pathological issues that lie outside the scope of needle diseases. Please see my comments below in the detailed review for suggested corrections.

This review serves to meet a pressing need for a current understanding of the potential applications of remote sensing to an emerging forest health threat in the southeast United States. The introduction serves as a concise review of BSNB for the reader and establishes enough understanding for the reader to see the utility of RS as a methodology to study the disease. The article is a good fit for PeerJ.

Experimental design

The study falls within the aims and scope of PeerJ, is thorough, and well described. A few issues with references and focused still exist throughout, and are highlighted in the detail review.

Validity of the findings

My primary outstanding apprehension lies in the framing PWD-related research. Please see the detail review below for my concerns with the inclusion of this data as currently presented.

Additional comments

Detail Review
Throughout: Make sure to double check the italicization of genera, species, and subspecies / forma specialis
Lines 43-44: Dothistroma needle blight may be caused by either D. septosporum or D. pini
L46-50: Needle diseases are caused by fungi that infect conifer foliage and elicit symptoms of disease in the needle tissues. These needle diseases are distinct from diseases caused by fungal infections of other tissues, for instance root rots, wilts, or cankers, where needles may be affected by these downstream effects but are not directly attacked. Neither pitch canker (a branch and stem canker) nor white pine blister rust (a branch and stem rust) are needle diseases. As such their mention is not relevant to this review. More appropriate alternatives worth mentioning are Swiss needle cast of Douglas fir (Nothophaeocryptopus geaumannii) or red needle cast of pines and Douglas fir (Phytophthora pluvialis).
L57: Theron et al (2022) is not an appropriate reference for this claim, as the authors make no mention of P. taeda infection by L. acicola. Instead, this study describes the infection of other Pinus species by L. pharomachri in Colombia. More appropriate references to the ongoing situation in the southeastern US may be found in the thesis by Datta (2021), Meinecke et al (2024 Phytobiomes), and in the recent extension materials by Meinecke et al (2024, WSFNR-24-25A, https://bugwoodcloud.org/resource/files/32519.pdf), Datta et al (2021, FOR-2105, https://www.aces.edu/blog/topics/forestry/brown-spot-needle-blight-of-loblolly-pine/) and Cunningham (2022, FSA5022, https://www.uaex.uada.edu/publications/PDF/FSA-5022.pdf).

L70-71: It would be helpful to expand on this somewhat and provide evidence, a reference to the current EPPO record of distribution and detections would suffice.
L79: Rather than pine decline, BSNB would be a more compelling emerging threat to cite. At this point it is widely recognized as a major issue and would remain consistent with the rest of the article.
L288-291: Since you afford some detail to describing the work of Janousek et al (2016), you may wish to expand somewhat on the work of Marcet-Houben et al (2024). Theirs is the first genome-wide population study on the species.
L451-473: Conceptually, the framing of this section on pine wilt disease is the most serious problem in the manuscript. Pine wilt disease is not a needle disease but instead is a wilt caused by the occlusion of the vasculature by the nematode causal agent. However, the findings may prove valuable if properly contextualized. Should this section be kept, it must be made clear that PWD is not a needle disease but instead a wilt disease system to which a great deal of RS-based research has been applied. For instance, it will be important to explicitly state how the inherent differences in disease progression and symptom presentation should be accounted for before applying the findings of PWD RS research directly to the case of BSNB.

Reviewer 3 ·

Basic reporting

Clarity and Language: I agree with the authors' response and believe that my recommendations have been implemented.

Literature Context and References: The authors added relevant references to the literature review, which significantly improved the quality of the manuscript.

The authors have revised the manuscript by breaking up long paragraphs to improve readability (lines 187-188, 412-413, 433-434). The authors have also carefully revised the figures and tables to ensure consistency in formatting, including the removal of italics from all figure titles. I agree with the authors' response and believe that my recommendations have been implemented.

Experimental design

The authors have really improved the transitions, especially between results and discussion, for a better reading experience. Also, the authors have increased the number of cross-references to figures and bibliometric maps, which ensured the perception of the material by readers. I agree with the authors' response and believe that my recommendations have been implemented.

Validity of the findings

The authors have indeed improved the analysis of the relevant literature. The revision of the conclusion has really strengthened the justification of the scientific significance of the study. The authors expanded on the discussion to include specific recommendations for future research. The discussed ground-based field validation improves the accuracy of remote sensing by providing reference data for model calibration and validation. The revised article notes that it provides reliable disease detection by linking spectral features to symptoms observed in the field, refining classification models and reducing misclassification caused by other stressors. I agree with the authors' response and believe that my recommendations have been implemented.

Additional comments

I recommend the article for publication

·

Basic reporting

no comment

Experimental design

no comment

Validity of the findings

no comment

Additional comments

I would like to thank the authors for the substantial revision of the review, it is a pleasure to read the improved text.

---

## Round 0.4 · accepted · Accept

Dear Doctor, I congratulate you on the acceptance of your article for publication. In the process of publication preparation, I ask you to correct two shortcomings on lines 35 and 330.

Reviewer 1 ·

Basic reporting

The manuscript has been extensively revised and added more contextual references in the introduction now as suggested to validate the need for this study.
The authors made necessary changes as suggested and the writing is easier to read and follow now.
There is no recent review for the specific disease as well

Experimental design

The survey methodology is refined now and the approach is better explained.

Validity of the findings

The findings are arranged in sub sections which allow the readers to follow the manuscript

Additional comments

The review is important considering the fact that there was no recent review available to assess the status of this disease.
Considering the significance of the disease, this manuscript will be helpful for many future directives.
Check line 330 for the order of citation.

Reviewer 2 ·

Basic reporting

Thank you for your thorough revisions. The current version meets the requirements set by PeerJ for satisfactory basic reporting. Please see a few last minor suggestions included below. Congratulations on this important and timely review.

Experimental design

Satisfactory, well done.

Validity of the findings

Upon review of the author's shift in framing, I feel the findings are now properly contextualized and valid.

Additional comments

Line 35: please refer to "fungal and oomycete pathogens" as Phytophthora pluvialis is an oomycete and not a fungus.

Great work, congratulations! I look forward to seeing this in print and to sharing with colleagues in the southeast US.